# Perinatal Outcome and Long-Term Infectious Morbidity of Offspring Born to Women with Known Tuberculosis

**DOI:** 10.3390/jcm9092768

**Published:** 2020-08-26

**Authors:** Shanny Sade, Tamar Wainstock, Eyal Sheiner, Gali Pariente

**Affiliations:** 1Department of Obstetrics and Gynecology, Soroka University Medical Center, Ben-Gurion University of the Negev, Beer-Sheva 8457108, Israel; sheiner@bgu.ac.il (E.S.); galipa@bgu.ac.il (G.P.); 2Department of Public Health, Faculty of Health Sciences, Ben-Gurion University of the Negev, Beer-Sheva 8457108, Israel; wainstoc@bgu.ac.il

**Keywords:** cesarean delivery, infectious morbidity, perinatal outcome, placental abruption, tuberculosis, very low birth weight

## Abstract

Objective: To evaluate the perinatal outcome of women with tuberculosis and to assess a possible association between maternal tuberculosis and long-term infectious morbidity of the offspring. Study design: Perinatal outcome and long-term infectious morbidity of offspring of mothers with and without tuberculosis were assessed. The study groups were followed until 18 years of age tracking infectious-related morbidity and infectious-related hospitalizations and then compared. For perinatal outcome, generalized estimation equation models were used. A Kaplan-Meier survival curve was used to compare cumulative incidence of long-term infectious morbidity. A Cox proportional hazards model was conducted to control for confounders. Results: During the study period, 243,682 deliveries were included, of which 46 (0.018%) occurred in women with tuberculosis. Maternal tuberculosis was found to be independently associated with placental abruption, cesarean deliveries, and very low birth weight. However, offspring born to mothers with tuberculosis did not demonstrate higher rates of infectious-related morbidity. Maternal tuberculosis was not noted as an independent risk factor for long-term infectious morbidity of the offspring. Conclusion: In our study, maternal tuberculosis was found to be independently associated with adverse perinatal outcomes. However, higher risk for long-term infectious morbidity of the offspring was not demonstrated. Careful surveillance of these women is required.

## 1. Introduction

Tuberculosis is a contagious disease caused by *Mycobacterium tuberculosis* and is the second most common infectious cause of death in adults worldwide [1]. The disease appeared hundreds of years ago and remained sporadic until the Industrial Revolution in the 18th century, when it became endemic, owing to the increased population density and unfavorable living conditions [1]. About one third of the world’s population is infected with tuberculosis, 75% of whom are 15–54 years old [2]. Immunosuppressed individuals are at risk of developing active tuberculosis once infected, particularly patients with Acquired Immuno-Deficiency Syndrome (AIDS) [1]. Despite a rigorous global effort over the past two decades to reduce the burden of tuberculosis by developing new drugs, diagnostics, and vaccines, it remains a global emergency. Tuberculosis is responsible for the deaths of more than 1 million people every year, most of them in low-income countries, with the majority of cases occurring in Asia (58%) and Africa (28%) [3,4,5].

Tuberculosis affects almost every organ in the body, and the most common site of the infection is the lungs [2]. In 15–20% of active cases, the infection spreads outside the lungs, causing other kinds of tuberculosis, collectively referred to as extrapulmonary tuberculosis [6]. Urogenital tuberculosis is the third most common form of extrapulmonary tuberculosis. It arises from hematogenous spread from a pulmonary or other non-genital source and most often involves the fallopian tubes, the endometrial cavity and the ovaries. Genital tract tuberculosis is associated with 21% of infertility cases in developing countries, due to tubal obstruction or adhesions in the uterine cavity [7].

Worldwide, approximately 500–800 million women are infected with tuberculosis, at least 216,000 during pregnancy [8]. It is a leading cause of death of women of child-bearing age (15–44 years) and, if untreated, a common cause of maternal and infant mortality [8,9]. Maternal tuberculosis has been described as a risk factor for adverse pregnancy outcomes including spontaneous abortions, preterm births, deliveries of low birth weight infants and higher rates of neonatal mortality [2,10,11,12,13,14].

Congenital tuberculosis is a rare complication of in utero tuberculosis infection, while the risk of postnatal transmission is significantly higher [2]. The transmission of congenital tuberculosis can be transplacental, or through the aspiration of the infected amniotic fluid [15]. The mortality rate is high in infants, especially in the absence of treatment. Early diagnosis is critical but challenging because of nonspecific symptoms [2,15].

Meconium-stained amniotic fluid has been associated with various pregnancy pathologies including microbial invasion of the amniotic cavity [16] and is also a well-established risk factor for immediate adverse neonatal outcomes, among which is neonatal sepsis [17,18]. The association between infectious processes (chorioamnionitis, neonatal sepsis) and meconium-stained amniotic fluid has raised a question regarding the presence of bacterial content in the meconium [19]. Nevertheless, some studies demonstrated lower rates of long-term infectious morbidity in offspring exposed to meconium-stained amniotic fluid and other intra-uterine inflammatory conditions [20,21].

It is possible that maternal tuberculosis during pregnancy also promotes a uterine environment that may affect the fetal future response to infections. Hence, in this study, we opted to evaluate the perinatal outcome of women with tuberculosis and to assess a possible association between maternal tuberculosis and long-term infectious morbidity of the offspring.

## 2. Materials and Methods

### 2.1. Study Design

A population-based retrospective cohort study was conducted. The study investigated perinatal outcomes of mothers with tuberculosis and the long-term infectious morbidity of offspring of mothers with tuberculosis. The primary exposure was maternal tuberculosis diagnosed before or during the current pregnancy. Any patient with relevant clinical manifestations (cough >2 to 3 weeks’ duration, lymphadenopathy, fevers, night sweats, weight loss) and relevant epidemiologic factors (history of prior tuberculosis infection or disease, known or possible tuberculosis exposure, and/or past or present residence in or travel to an area where tuberculosis is endemic), and/or positive Tuberculin Mantoux test, was administered as one of the diagnostic tests of tuberculosis. The possible diagnostic tests included were isolating mycobacterium tuberculosis from bodily secretions (culture of sputum, bronchoalveolar lavage, or pleural fluid), or tissue (pleural biopsy or lung biopsy) and sputum acid-fast bacilli smear, and nucleic acid amplification testing was also considered diagnostic for tuberculosis [22,23]. Offspring of mothers without tuberculosis comprised the comparison (unexposed) group.

Pregnancy characteristics, perinatal outcomes, and long-term infectious morbidity of the offspring were compared between offspring of mothers with and without tuberculosis. Perinatal outcomes such as maternal diabetes mellitus, hypertensive disorders, placental abruption, mode of delivery, very low birth weight, low Apgar scores and perinatal mortality were assessed. Infectious morbidity assessment included hospitalizations of the offspring up to the age of 18 years and was defined as any of the following diagnoses: urinary tract infections, gastroenteritis, meningitis, sepsis, pneumonia, viral infections, bacterial infections and fungal infections. The predefined ICD-9 code list of all diagnoses included in each of these conditions is detailed in the Appendix A. Follow-up time was defined as time until an event (hospitalization with any infectious diagnosis). Follow-up was terminated if any of the following occurred: first hospitalization with any infectious diagnosis (i.e., an event), hospitalization resulting in death, or when the child reached 18 years of age.

### 2.2. Definitions

Maternal tuberculosis was defined as maternal tuberculosis during pregnancy or personal history of tuberculosis (ICD-9 codes 64731, V1201). Maternal diabetes was defined as pre-gestational diabetes and gestational diabetes type A1 (not medically treated) and A2 (medically treated). Hypertensive disorders were defined as chronic hypertension, gestational hypertension and preeclampsia. Very low birth weight was defined as a birth weight of less than 1500 g. Low Apgar scores were defined as an Apgar score lower than 7 at 1 min or 5 min. Perinatal mortality was defined as intra-uterine fetal death, intrapartum death, or post-partum death.

### 2.3. Settings and Study Population

The study was conducted at the Soroka University Medical Center, a tertiary medical center and the only hospital in the southern region of the country, which occupies 65% of Israel’s territory (approximately 1.3 million people) [24]. The institutional review board, in accordance with the Helsinki declaration, approved the study (IRB number 0357-19-SOR).

All singleton pregnancies of women who delivered between the years 1991 and 2014 at the Soroka University Medical Center were included in the study. Multiple pregnancy and children with congenital malformations or chromosomal abnormalities were excluded from the study. Cases of perinatal deaths were excluded from the long-term analysis.

### 2.4. Data Collection Method

Data were collected from two databases that were cross-linked and merged, based on mothers’ and infants’ identification numbers: the computerized perinatal database of the Obstetrics and Gynecology department and the pediatric hospitalization database (“Demog-ICD9”). The obstetrical database includes demographic information, perinatal assessments, maternal morbidities and maternal and fetal outcomes and is recorded immediately following delivery by the attending physician. Records were anonymized prior to analysis. The pediatric hospitalization database includes demographic information and international classification of diseases, ninth revision codes (ICD-9), for all medical diagnoses made during any hospitalization. All newborns in the country are issued with a national security number (ID number) which is then registered in the mother’s formal identification card. These identification numbers are not changed nor duplicated within the population at any given time. This allows certainty of the relationship between any mother and child in our datasets.

### 2.5. Statistical Analysis

Univariable analysis was performed to compare background characteristics between the two study groups. The univariable analysis included Chi-square tests for categorical variables, and t-tests or Mann–Whitney U tests for continuous variables according to their distribution. To compare perinatal outcomes, generalized estimation equation models were used to control for confounders. A binary logistic regression model was used to study the association between tuberculosis exposure and perinatal outcomes. The Generalized Estimation Equation binary logistic model allows adjustments for siblings in the cohort and the similarities between them. Cumulative incidence rates of infectious-related hospitalizations were compared using Kaplan–Meier survival curves, and the log-rank test was used to determine significant differences. A Cox proportional hazards model was conducted to control for confounders. All analysis was performed using SPSS package 23.0, IBM SPSS statistics for Windows, version 23.0. Armonk, NY, USA: IBM Corp as well as the STATA software 12th ed, stataCorp. Stata Statistical Software: Release 12. College Station, TX, USA: StataCorp LP. According to previous studies, the incidence of cesarean delivery and very low birth weight in the general population is ~21.1% and 0.6% of all deliveries, respectively [25,26]. Assuming the rate in the unexposed group is equivalent to that of the entire population and given α = 0.05, for an odds ratio of 2.513 and 7.540, the calculated power is 71% and 52% for cesarean delivery and very low birth weight, respectively.

## 3. Results

During the study period, 243,682 singleton deliveries met the inclusion criteria, of which 46 (0.018%) offspring were born to mothers with tuberculosis.

Our study included 10 women with active tuberculosis, diagnosed during the pregnancy. All had pulmonary tuberculosis and were treated with combination of antibiotic therapy such as isoniazid, azithromycin, ethambutol and pyrazinamide. The other 36 women had latent tuberculosis. No cases of genital tuberculosis were seen in our population.

Demographic characteristics of the study groups are presented in Table 1. Of the 46 women with tuberculosis, 2 women were also HIV carriers. These women had active tuberculosis during pregnancy.

Table 2 summarizes the perinatal outcomes of both groups.

Due to the relatively small number of cases of tuberculosis, the search was extended until December 2019. One additional case of tuberculosis was found. The additional patient, who had latent tuberculosis delivered preterm (29 weeks gestation), a very low birth weight neonate, which substantiates the significant association found in our study between tuberculosis and delivery of very low birth weight neonates.

There were three cases of placental abruption, of which one was a case of perinatal mortality. The only perinatal mortality was an ante-partum death case at term in a woman with latent tuberculosis.

Significantly higher rates of very low birth weight newborns were demonstrated in women with latent tuberculosis when compared to women with active tuberculosis during pregnancy, latent tuberculosis, and women without tuberculosis (0.0%, 5.6%, and 0.6%, respectively, *p* < 0.001). In addition, significantly higher rates of cesarean deliveries were demonstrated in all women with tuberculosis, when compared to women with active tuberculosis during pregnancy, latent tuberculosis, and women without tuberculosis (30%, 27%, and 13.6%, respectively, *p* = 0.014). No differences in rates of very low birth weight and cesarean delivery were demonstrated between active and latent tuberculosis during pregnancy. Table 3 summarizes the indications of cesarean delivery of both groups of women in our study. Other than the rate of cesarean section due to placental abruption, no differences in the cesarean delivery indications were demonstrated between women with and without tuberculosis.

Using generalized estimation equation models, controlling for maternal age and gestational age separately, tuberculosis was found to be an independent risk factor for placental abruption and very low birth weight. The association between maternal tuberculosis and cesarean delivery remained independent when controlling for gestational age (OR: 2.085, 95% CI 1.14–3.80, *p* = 0.017), but lost its significance after controlling for maternal age (OR: 1.75, 95% CI 0.93–3.26, *p* = 0.078, Table 4).

After excluding all cases of perinatal mortality, the study population included 242,342 offspring, among them were 45 offspring to mothers with tuberculosis. All children were retrieved for follow-up except for one case of ante-partum fetal death.

No significant differences in total infectious hospitalization rates were noted between offspring born to mothers with and without tuberculosis (6.7 vs. 10.0%, *p* = 0.621). Likewise, the Kaplan–Meier survival curve did not demonstrate higher cumulative incidence of infectious morbidity among offspring to mothers with tuberculosis (Log-rank test *p* = 0.423, Figure 1.). Using a Cox proportional hazards model, controlled for gestational age at birth, maternal tuberculosis was not independently associated with long-term infectious morbidity of the offspring (adjusted HR 0.6, 95% CI 0.19–1.89, *p* = 0.395).

No differences in rates of long-term infectious morbidity of the offspring were demonstrated among women with active tuberculosis during pregnancy, latent tuberculosis, and women without tuberculosis (0.0%, 8.3%, and 10%, respectively, *p* = 0.574). No difference in rate of long-term infectious morbidity of the offspring was demonstrated between active and latent tuberculosis during pregnancy.

## 4. Discussion

In this large population-based cohort study, we found that placental abruption, cesarean delivery and very low birth weight neonates were all more frequent in mothers with tuberculosis compared to mothers without tuberculosis. However, no significant difference in long-term infectious morbidity was demonstrated between offspring of mothers with and without tuberculosis.

### 4.1. Short Term Perinatal Outcomes of Women with Tuberculosis

Maternal tuberculosis was previously noted as a risk factor for adverse short-term perinatal outcome. Our study found that placental abruption was independently associated with maternal tuberculosis. In contrast to our results, El-Messidy et al. did not observe an increased risk of placental abruption or other placental pathology, such as placenta previa [9]. Differences in the diagnostic criteria for placental abruption may explain these differences.

In our study, the association between maternal tuberculosis and cesarean delivery remained independent when controlling for gestational age but lost its significance after controlling for maternal age. A controversy exists in the literature regarding the risk of cesarean delivery among women with tuberculosis. While Sobhy et al. in their systematic review demonstrated an association between active tuberculosis during pregnancy and higher risk for cesarean delivery [12], Yadav et. al., who investigated only women with extra-pulmonary tuberculosis before or during pregnancy, failed to demonstrate such an association [5]. Differences in population size and disease nature may explain these dissimilarities seen among the different studies.

The association between maternal tuberculosis and delivery of small for gestational age or lower birth weight infants was established in previous studies [2,5,11,12,13,27]. Although low birth weight can be attributed to fetal growth retardation [28], in most studies it is explained by higher rates of prematurity seen among women with tuberculosis [2,5,12,29]. Our study found independent association between maternal tuberculosis and delivery of a very low birth weight infant, regardless of gestational age. Moreover, our study failed to demonstrate differences in rates of preterm delivery between mothers with or without tuberculosis. This lack of association may be explained by differences in medical care provided in low resource populations compared to countries in which healthcare is universal and enables equal and free medical care to all citizens, regardless of their socioeconomic status.

Despite previous studies showing strong association between genital tract tuberculosis and infertility [7], our study did not demonstrate such an association, although in vitro fertilization procedures in Israel are not performed in women with active tuberculosis.

### 4.2. Long- Term Outcomes of Offspring Born to Women with Tuberculosis

In this large population-based cohort study of nearly 250,000 deliveries, with a follow up period of more than 20 years, no differences in long-term infectious morbidity were demonstrated between offspring to women with and without tuberculosis. To the best of our knowledge, our study is the first to investigate long-term morbidity of offspring born to mothers with tuberculosis.

Previous studies associated maternal inflammatory statuses with offspring with long-term alterations of immune responses that result in childhood infectious morbidity. These maternal conditions were pure infectious diseases [19,30,31], but were also attributed to conditions such as obesity and smoking that may involve maternal subclinical inflammation [20,21,32,33,34]. This association between intrauterine inflammation and childhood morbidity, however, showed mixed trends in different studies. Melville et al. demonstrated that infants exposed to intrauterine inflammation show increased immune maturation and require less intensive and respiratory care after birth [20]. Likewise, exposure to meconium-stained amniotic fluid was associated with lower incidence of long-term infectious morbidity of the offspring [21]. In contrast, other studies showed increased risk of short and long- term infectious morbidity in children born to mothers with an augmented inflammatory state during pregnancy. Preterm birth, maternal urinary tract infection, maternal obesity and maternal substance abuse were all demonstrated to be associated with increased infectious morbidity of the offspring [30,31,32,33,34,35]. We hypothesized that maternal tuberculosis can also promote an intrauterine environment that may alter immune responses of the offspring in his future. The lack of association between maternal tuberculosis and long-term infectious morbidity of the offspring seen in our study can be explained by the low prevalence of tuberculosis in our population. Another explanation relates to the timing of the tuberculosis infection. If the acute tuberculosis infection had occurred and was treated before pregnancy, the maternal inflammatory response may have been attenuated during pregnancy and carried less effect on the exposed fetus. Finally, the lack of association might be due to the fact that some long-term infectious-related morbidities only manifest at ages older than 18 years. 

### 4.3. Strength and Limitations of the Study

Our study’s main strength stems from its population-based nature, utilizing a large cohort, with a long follow-up period. Our hospital is the only tertiary hospital serving the entire population of the region. The hospital provides both maternal and pediatric services, and hence enables us to combine databases from obstetrical and pediatric departments, from pregnancy to adulthood. These facts allow for a low probability of selection bias. This large cohort enables a statistical power, while controlling for certain parameters during pregnancy and delivery.

However, our study has several limitations. The main limitation is the study’s retrospective design, which naturally suggests association but not necessarily causation between maternal tuberculosis, perinatal outcome, and long- term childhood infectious morbidity. The long follow-up period of this study is also a limitation since management and screening of the diseases have changed over the course of the study. Another limitation of the study is related to the low number of mothers with tuberculosis, which may limit the statistical interpretation concerning outcome and outcome- associated factors. Extending the timeline of the study to capture more patients or conducting the same study in other regions with higher incidence might have strengthened the results of this study, as the significant finding affected only a low number of pregnancies. Further studies using a multi-center database analysis should be done in order to further investigate perinatal outcome of women with known tuberculosis. The significant findings of increased abruption and very low birthweight in mothers with tuberculosis only affected three and two pregnancies, respectively. Nevertheless, it is noteworthy that the results remain significant even after controlling for maternal age and gestational age. Although the statistics support there being a difference, rationally this may still be due to chance alone. In addition, although Soroka University Medical Center is the sole medical center in the area, and is a tertiary and a referral center, data from a single center does not guarantee that all children of mothers with tuberculosis and all their health problems were included in the study. Lastly, since there is no general screening in Israel for tuberculosis, differential misclassification of the study exposure (cases with tuberculosis) may have appeared in the comparison group (unexposed group), meaning that among the unexposed group, there were actually undiagnosed cases of tuberculosis. This, however, may have led to an underestimation of the true association between tuberculosis with adverse perinatal outcome found in our study.

## 5. Conclusions

In conclusion, maternal tuberculosis is an independent risk factor for placental abruption, cesarean delivery and very low birth weight. However, in our population, maternal tuberculosis did not appear to have a significant impact on long term infectious morbidity of the offspring. Further studies should investigate the association between maternal tuberculosis and other related morbidities of the offspring, among different cohorts, in order to identify populations at risk for long- term morbidity.

## Figures and Tables

**Figure 1 jcm-09-02768-f001:**
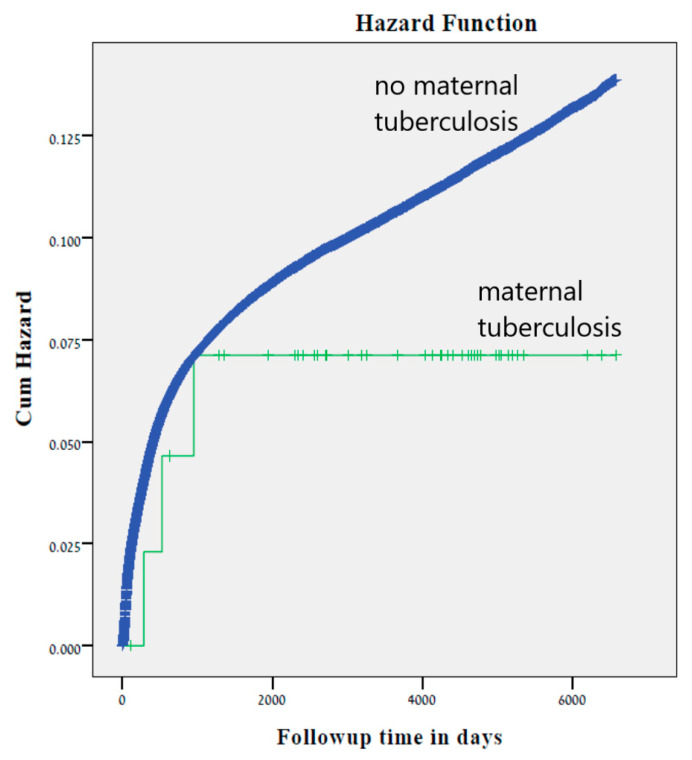
Kaplan Meier survival curve demonstrating the cumulative incidence of infectious hospitalizations according to pregnancy with and without maternal tuberculosis (Log-Rank test *p* = 0.423).

**Table 1 jcm-09-02768-t001:** Demographic characteristics of the study population.

Variables		Mother with Tuberculosis (*n* = 46)	Mother Without Tuberculosis (*n* = 243,636)	*p* Value
Maternal Age, Years (Mean ± SD)		30.5 ± 6.4	28.1 ± 5.8	0.01
Gravidity (%)	1	10.9	19.7	0.132
	2+	89.1	80.3	
Parity (%)	1	13.0	23.6	0.092
	2+	87.0	76.4	
Pregnancy Conceived After In-Vitro Fertilization (%)		2.2	1.1	0.65
Obesity (%)		0.0	1.0	0.491
Chronic Hypertension (%)		4.3	1.4	0.080
Diabetes Mellitus (%)		4.3	0.7	0.004

**Table 2 jcm-09-02768-t002:** The association between maternal tuberculosis and adverse perinatal outcome; univariable analysis.

Characteristic		Mother with Tuberculosis (*n* = 46) *n* (%)	Mother without Tuberculosis (*n* = 243,636) *n* (%)	*p* Value
Gestational Diabetes Mellitus		3 (6.5)	10,388 (4.3)	0.449
Preeclamsia/Eclampsia		3 (6.5)	9610 (3.9)	0.369
Placenta Previa		0 (0.0)	934 (0.4)	1.00
Placental Abruption		3 (6.5)	1356 (0.6)	0.002
Preterm Delivery	Before 37 weeks gestation	4 (8.7)	16,716 (6.9)	0.55
	Before 34 weeks gestation	2 (4.3)	3307 (1.4)	0.13
Mode of Delivery	Vaginal delivery	32 (69.6)	202,816 (83.2)	0.01
	Assisted delivery	1 (2.2)	7807 (3.2)	
	Cesarean delivery	13 (28.3)	33,013 (13.6)	
Very Low Birth Weight		2 (4.3)	1460 (0.6)	0.03
Low Apgar Score At 1st Minute (<7)		4 (8.7)	12,986 (5.3)	0.31
Low Apgar Score At 5th Minute (<7)		1 (2.2)	5508 (2.3)	>0.99
Perinatal Mortality		1 (2.2)	1339 (0.5)	0.22

**Table 3 jcm-09-02768-t003:** Operative indications of women with and without tuberculosis *.

	Mother with Tuberculosis (%)	Mother without Tuberculosis (%)	*p* Value
Transverse Lie	0.0	3.3	0.509
Face/Braw Presentation	0.0	0.8	0.746
Compound Presentation	0.0	0.5	0.804
Complete Breech	0.0	12.6	0.170
Footling Presentation	0.0	6.9	0.327
Cephalo-Pelvic Disproportion	7.7	1.5	0.061
Arrest of Dilatation	0.0	12.4	0.174
Arrest of Descent	0.0	4.2	0.451
Placental Abruption	23.1	2.9	<0.001
Placenta Previa	0.0	2.7	0.551
Prolapse of Cord	7.7	2.1	0.157
Non-Reassuring Fetal Heart Rate	7.7	5.7	0.759

* Some cesarean section had more than one indication, therefore the sum is over 100%.

**Table 4 jcm-09-02768-t004:** Generalized estimation equation models for placental abruption, cesarean delivery, and very low birth weight, controlling for maternal age and gestational age.

	Outcome	Controlled For	OR 95% CI	*p* Value
**1.**	Placental Abruption	Maternal Age	10.76 (3.37–34.36)	0.002
		Gestational Age	11.27 (3.44–36.96)	<0.001
**2.**	Cesarean Delivery	Maternal Age	1.75 (0.93–3.26)	0.078
		Gestational Age	2.085 (1.14–3.80)	0.017
**3.**	Very Low Birth Weight	Maternal Age	6.91 (1.77–26.95)	0.005
		Gestational Age	7.37 (2.05–26.42)	0.002

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
