# Peer review of "Perinatal Outcome and Long-Term Infectious Morbidity of Offspring Born to Women with Known Tuberculosis"

_jcm, 2020, doi:10.3390/jcm9092768_

Round 1

Reviewer 1 Report

Thank you very much for the revised version of your manuscript. 

Author Response

Referee 1:

Thank you very much for the revised version of your manuscript.

Response: We thank the reviewer for this comment

We are very grateful for the provided feedback and feel that these revisions had considerably improved the quality of this paper. It is our sincere hope that those revisions will be met by approval of the Editorial Board.

Respectfully,

Shanny Sade MD, Eyal Sheiner MD PhD, Tamar Wainstock PhD, Narkis Hermon MD, Shimrit Yaniv Salem MD, Tamar Kosef MD, Talya Lanxner Battat MD, Sharon Oron MD, Gali Pariente MD

Reviewer 2 Report

The paper concerns an interesting subject, however it needs major revision.

The manuscript design is based on incorrect assumptions. The study investigated perinatal outcomes of mothers with tuberculosis diagnosed by a Tuberculin Mantoux test. The test is used as a screening tool, however its specificity is too low to be a diagnostic tool. In fact most positive reactions in low-risk individuals are false positives. Therefore it needs another diagnostic tool to establish a diagnosis of tuberculosis. False positive results may be caused by nontuberculous mycobacteria, allergic reaction or hypersensitivity, or even when the injected area is touched, causing swelling and itching. On the other hand the test may be false negative in case of recent TB infection (less than 8–10 weeks), mononucleosis, live virus vaccine within the last 3 weeks, sarcoidosis, Hodgkin’s disease, steroid use, malnutricion or ever upper respiratory virus infection. The authors should refine diagnostic criteria in the study and positive Tuberculin test is not synonymous with tuberculosis.

Therefore the article title “….. with known tuberculosis” is not appropriate.

The conclusion part of the abstract is actually repeating of the results and should be modified.

In the introduction part line 51 – “Worldwide, approximately 500-800 million women are infected with tuberculosis, at least 216,000 during pregnancy.” – please provide a reference for this data.

Line 64 – “While some studies demonstrated lower rates of long-term infectious morbidity in offspring exposed to meconium stained amniotic fluid and other intra-uterine inflammatory conditions, (16,17)…” – explain why do you consider meconium stayed amniotic fluid an inflammatory condition?

In the study methodology the authors should revise which of the mothers were actually diagnosed with tuberculosis and which had false positive test. What was the time period of tuberculosis before the pregnancy – specify this information in the inclusion criteria. Acute tuberculosis during pregnancy and  that the one diagnosed long before the pregnancy should not be included in the same study group. Where the mothers given specific treatment for tuberculosis or not? Is there a relation between perinatal outcome and treatment?

Data from a single centre does not guarantee that all children of mothers with tuberculosis and all their health problems were included in the study. It should be mentioned in the study limitations.

In the results section the authors should avoid providing the same data in the text and in the table. Therefore below sentences should be modified or removed:

“Mothers with tuberculosis were older with higher gravidity and parity order. No differences were found in the in-vitro fertilization rates between the two groups.”;

“Mothers with tuberculosis had significantly higher rates of placental abruption (6.5 vs. 0.6%, p=0.002), caesarean delivery (28.3 vs. 13913.6%, p=0.01) and very low birth weight (4.3 vs. 0.6%, p=0.03).”’

“Rates of maternal diabetes mellitus and hypertensive disorders were comparable between the groups. Likewise, neonatal outcomes such as low Apgar scores at 1stand 5thminutes and rates of small for gestational age neonates were comparable between the groups.”

Table 1 should be extended with information on maternal BMI and chronic diseases, especially HIV.

The authors found significantly more cesarean deliveries in a group of women with positive Mantoux test. Please provide indications for cesarean sections.

To verify if positive Mantoux test is an independent risk factor for perinatal complications a logistic regression analysis should be performed. Please provide it.

Please provide a necessary number of study group individuals on the basis of power analysis.

The authors found a relation between positive Mantoux test and placental abruption as well as LBW. What is the hypothetic etiology of these relations? Please provide them in the discussion part of the study.

Line 206 – “Despite previous studies showing strong association between genital tract tuberculosis and infertility (7), our study did not demonstrate such an association…” – The data on genital tract tuberculosis occurrence in the study group is missing – please provide specific data.

The English language and space inserting in the text need throughout editing.

Author Response

Referee 2:

  1. The manuscript design is based on incorrect assumptions. The study investigated perinatal outcomes of mothers with tuberculosis diagnosed by a Tuberculin Mantoux test. The test is used as a screening tool; however, its specificity is too low to be a diagnostic tool. In fact, most positive reactions in low-risk individuals are false positives. Therefore, it needs another diagnostic tool to establish a diagnosis of tuberculosis. False positive results may be caused by nontuberculous mycobacteria, allergic reaction or hypersensitivity, or even when the injected area is touched, causing swelling and itching. On the other hand, the test may be false negative in case of recent TB infection (less than 8–10 weeks), mononucleosis, live virus vaccine within the last 3 weeks, sarcoidosis, Hodgkin’s disease, steroid use, malnutrition or ever upper respiratory virus infection. The authors should refine diagnostic criteria in the study and positive Tuberculin test is not synonymous with tuberculosis. Therefore, the article title “…. with known tuberculosis” is not appropriate.

Response: We thank the reviewer for the comment. We have consulted with the infectious disease department of our institution regarding the diagnostic tools of tuberculosis of our cohort and opened several files for verification. Accordingly, we have revised the definition of women with tuberculosis. Any patient with relevant clinical manifestations (cough >2 to 3 weeks' duration, lymphadenopathy, fevers, night sweats, weight loss) and relevant epidemiologic factors (history of prior tuberculosis infection or disease, known or possible tuberculosis exposure, and/or past or present residence in or travel to an area where tuberculosis is endemic), and/or positive Tuberculin Mantoux test, was administered one of the diagnostic tests of tuberculosis. The possible diagnostic tests included isolating mycobacterium tuberculosis from bodily secretions (culture of sputum, bronchoalveolar lavage, or pleural fluid), or tissue (pleural biopsy or lung biopsy) and sputum acid-fast bacilli smear and nucleic acid amplification testing.

The following paragraph has been added to the Materials and Methods section: "The primary exposure was maternal tuberculosis diagnosed before or during the current pregnancy. Any patient with relevant clinical manifestations (cough >2 to 3 weeks' duration, lymphadenopathy, fevers, night sweats, weight loss) and relevant epidemiologic factors (history of prior tuberculosis infection or disease, known or possible tuberculosis exposure, and/or past or present residence in or travel to an area where tuberculosis is endemic), and/or positive Tuberculin Mantoux test, was administered one of the diagnostic tests of tuberculosis. The possible diagnostic tests included were isolating mycobacterium tuberculosis from bodily secretions (culture of sputum, bronchoalveolar lavage, or pleural fluid), or tissue (pleural biopsy or lung biopsy) and sputum acid-fast bacilli smear and nucleic acid amplification testing were also considered diagnostic for tuberculosis (22,23)".

            References:

  1. Lewinsohn DM, Leonard MK, Lobue PA et al. Official American thoracic society/infectious diseases society of America/centers for disease control and prevention clinical practice guidelines: diagnosis of Tuberculosis in adults and children. Clin Infect Dis. 2017;15;64(2):111-115.
  2. Pai M, Nicol MP, Boehme CC. Tuberculosis diagnostics: state of the art and future directions. Microbiol Spectr. 2016;4(5).

           (Materials and Methods section, lines 90-99 in the revised manuscript)

  1. The conclusion part of the abstract is actually repeating of the results and should be modified.

Response: The conclusion part of the abstract has been revised as follows: "In our study, maternal tuberculosis was found to be independently associated with adverse perinatal outcomes. However, higher risk for long term infectious morbidity of the offspring was not demonstrated. Careful surveillance of these women is required".

(Abstract section, lines 25-27 in the revised manuscript)

  1. In the introduction part line 51 – “Worldwide, approximately 500-800 million women are infected with tuberculosis, at least 216,000 during pregnancy.” – please provide a reference for this data.

Response: The following reference has been provided at the end of the sentence.

Reference number 8: Gupta A, Mathad JS, Abdel-Rahman SM et al. Toward earlier inclusion of pregnant and postpartum women in tuberculosis drug trials: consensus statements from an international expert panel. Clin Infect Dis. 2016;62:761-769.

(Introduction section, line 55 in the revised manuscript)

  1. Line 64 – “While some studies demonstrated lower rates of long-term infectious morbidity in offspring exposed to meconium stained amniotic fluid and other intra-uterine inflammatory conditions, (16,17) …” – explain why do you consider meconium stained amniotic fluid an inflammatory condition?

Response:

The following sentence has been added to the Introduction section: "Meconium- stained amniotic fluid has been associated with various pregnancy pathologies including microbial invasion of the amniotic cavity (16) and is also a well‐established risk factor for immediate adverse neonatal outcomes among them is neonatal sepsis (17,18). The association between infectious processes (chorioamnionitis, neonatal sepsis) and meconium- stained amniotic fluid has raised a question regarding the presence of bacterial content in the meconium (19). Nevertheless, some studies demonstrated lower rates of long-term infectious morbidity in offspring exposed to meconium stained amniotic fluid and other intra-uterine inflammatory conditions (20,21) ".

References:

  1. Romero R, Yoon BH, Chaemsaithong P et al. Bacteria and endotoxin in meconium-stained amniotic fluid at term: could intra-amniotic infection cause meconium passage? J Matern Fetal Neonatal Med. 2014;27(8):775-88.
  2. Brabbing-Goldsein D, Nir D, Cohen D, Many A, Malovitz S. Preterm meconium-stained amniotic fluid is an ominous sign for the development of chorioamnionitis and for in utero cord compression. J Matern Fetal Neonatal Med. 2017;30(17):2042-2045.
  3. Sheiner E, Hadar A, Shoham-Vardi I, Hallak M, Katz M, Mazor M. The effect of meconium on perinatal outcome: a prospective analysis. J Matern Fetal Neonatal Med. 2002;11(1):54-9.
  4. Tapiainen T, Paalanne N, Tejesvi MV et al. Maternal influence on the fetal microbiome in a population-based study of the first-pass meconium. Pediatr Res. 2018;84(3):371-379.

(Introduction section, lines 65-71 in the revised manuscript)

  1. In the study methodology the authors should revise which of the mothers were actually diagnosed with tuberculosis and which had false positive test. What was the time period of tuberculosis before the pregnancy – specify this information in the inclusion criteria. Acute tuberculosis during pregnancy and that the one diagnosed long before the pregnancy should not be included in the same study group. Where the mothers given specific treatment for tuberculosis or not? Is there a relation between perinatal outcome and treatment?

Response: We thank the reviewer for the comment. The following paragraph has been added to the Results section: "Our study included 10 women with active tuberculosis, diagnosed during the pregnancy. All had pulmonary tuberculosis and were treated with combination of antibiotic therapy such as Isoniazid, Azithromycin, Ethambutol and Pyrazinamide.  The other 36 women had latent tuberculosis".

(Results section, lines 162-165 in the revised manuscript)

Perinatal outcome and long-term infectious morbidity of the offspring according to the disease status during pregnancy are described in the following table: 

Mother with active tuberculosis (n=10)

(%)

Mother with latent tuberculosis (n=36)

(%)

Mother without tuberculosis (n=243,636)

(%)

P value

Very low birth weight

0.0

5.6

0.6

<0.001

Cesarean delivery

30

27

13.6

0.014

Long-term infectious morbidity of the offspring

0.0

8.3

10

0.574

The following paragraph has been added to the Results section:

"Significantly higher rates of very low birth weight newborns were demonstrated in women with latent tuberculosis when compared in women with active tuberculosis during pregnancy, latent tuberculosis and women without tuberculosis (0.0%, 5.6% and 0.6%, respectively, p <0.001). Also, significantly higher rates of cesarean deliveries were demonstrated in all women with tuberculosis, when compared in women with active tuberculosis during pregnancy, latent tuberculosis and women without tuberculosis (30%, 27% and 13.6%, respectively, P= 0.014)".

(Results section, lines 190-195 in the revised manuscript)

The following paragraph has been added to the Results section:

"No differences in rates of long-term infectious morbidity of the offspring were demonstrated among women with active tuberculosis during pregnancy, latent tuberculosis and women without tuberculosis (0.0%, 8.3% and 10%, respectively, p=0.574)".

(Results section, lines 227-229 in the revised manuscript)

We further compared perinatal outcome and long-term infectious morbidity of the offspring according to disease activity during pregnancy. The results are described in the following table:

Mother with active tuberculosis (n=10)

(%)

Mother with latent tuberculosis (n=36)

(%)

P value

Very low birth weight

0.0

5.6

0.446

Cesarean delivery

30

27.8

0.890

Long-term infectious morbidity of the offspring

0.0

8.3

0.370

The following sentence has been added to the Results section:

"No differences in rates of very low birth weight and cesarean delivery were demonstrated between active and latent tuberculosis during pregnancy".

(Results section, lines 195-197 in the revised manuscript).

The following sentence has been added to the Results section: "No difference in rate of long-term infectious morbidity of the offspring was demonstrated between active and latent tuberculosis during pregnancy".

(Results section, lines 230-231 in the revised manuscript)

  1. Data from a single center does not guarantee that all children of mothers with tuberculosis and all their health problems were included in the study. It should be mentioned in the study limitations.

Response: The following paragraph has been added to the limitation section of the Discussion: "In addition, although Soroka University Medical Center is the sole medical center in the area, and is a tertiary and a referral center, data from a single center does not guarantee that all children of mothers with tuberculosis and all their health problems were included in the study".

(Discussion section, lines 314-317 in the revised manuscript)

  1. In the results section the authors should avoid providing the same data in the text and in the table. Therefore, below sentences should be modified or removed:

“Mothers with tuberculosis were older with higher gravidity and parity       order. No differences were found in the in-vitro fertilization rates between the two groups.”;

“Mothers with tuberculosis had significantly higher rates of placental    abruption (6.5 vs. 0.6%, p=0.002), caesarean delivery (28.3 vs. 13913.6%, p=0.01) and very low birth weight (4.3 vs. 0.6%, p=0.03).”’

“Rates of maternal diabetes mellitus and hypertensive disorders were comparable between the groups. Likewise, neonatal outcomes such as low Apgar scores at 1stand 5thminutes and rates of small for gestational age neonates were comparable between the groups.”

Response: The above sentences were omitted from the revised manuscript according to the comment of the reviewer.

(Results section, lines 168-169, 173-175, 181-183 in the revised manuscript)

  1. Table 1 should be extended with information on maternal BMI and chronic diseases, especially HIV.

Response: Table 1 has been revised according to the comment of the reviewer:

Variables

Mother with tuberculosis (n=46)

Mother without tuberculosis (n=243,636)

P value

Maternal age, years (mean+ SD)

30.5+ 6.4

28.1+ 5.8

0.01

Gravidity (%)

1

10.9

19.7

0.132

2+

89.1

80.3

Parity (%)

1

13.0

23.6

0.092

2+

87.0

76.4

Pregnancy conceived after in-vitro fertilization (%)

2.2

1.1

0.65

Obesity (%)

0.0

1.0

0.491

Chronic hypertension (%)

4.3

1.4

0.080

Diabetes mellitus (%)

4.3

0.7

0.004

(Results section, line 171 in the revised manuscript)

The following sentence has been added to the Results section: "Of the 46 women with tuberculosis, 2 women were also HIV carriers. These women had active tuberculosis during pregnancy".

(Results section, lines 166-168 in the revised manuscript)

Table 2 has been revised accordingly:

Characteristic

Mother with tuberculosis (n=46)

n (%)

Mother without tuberculosis (n=243,636)

n (%)

P value

Gestational diabetes mellitus

3(6.5)

10,388(4.3)

0.449

Preeclampsia/

eclampsia

3(6.5)

9,610(3.9)

0.369

Placenta previa

0(0.0)

934(0.4)

1.00

Placental abruption

3(6.5)

1,356(0.6)

0.002

Preterm delivery

Before 37 weeks gestation

4(8.7)

16,716(6.9)

0.55

Before 34 weeks gestation

2(4.3)

3,307(1.4)

0.13

Mode of delivery

Vaginal delivery

32(69.6)

202,816(83.2)

0.01

Assisted delivery

1(2.2)

7,807(3.2)

Cesarean delivery

13(28.3)

33,013(13.6)

Very Low birth weight

2(4.3)

1,460(0.6)

0.03

Low Apgar score at 1st minute (<7)

4(8.7)

12,986(5.3)

0.31

Low Apgar score at 5th minute (<7)

1(2.2)

5,508(2.3)

>0.99

Perinatal mortality

1(2.2)

1,339(0.5)

0.22

(Results section, lines 186-187 in the revised manuscript)

  1. The authors found significantly more cesarean deliveries in a group of women with positive Mantoux test. Please provide indications for cesarean sections.

Response: According to the comment of the reviewer, the following table and description have been added to the Results section:

"Table 3 summarizes the indications of cesarean delivery of both groups of women in our study. Other than the rate of cesarean section due to placental abruption, no differences in the cesarean delivery indications were demonstrated between women with and without tuberculosis.

Table 3: Operative indications of women with and without tuberculosis.

Mother with tuberculosis

(%)

Mother without tuberculosis

(%)

P value

Transverse lie

0.0

3.3

0.509

Face/braw presentation

0.0

0.8

0.746

Compound presentation

0.0

0.5

0.804

Complete breech

0.0

12.6

0.170

Footling presentation

0.0

6.9

0.327

Cephalo-pelvic disproportion

7.7

1.5

0.061

Arrest of dilatation

0.0

12.4

0.174

Arrest of descent

0.0

4.2

0.451

Placental abruption

23.1

2.9

<0.001

Placenta previa

0.0

2.7

0.551

Prolapse of cord

7.7

2.1

0.157

Non- reassuring fetal heart rate

7.7

5.7

0.759

*Some cesarean section had more than one indication, therefore, the sum is over 100%".

(Results section, lines 198-202 in the revised manuscript)

  1. To verify if positive Mantoux test is an independent risk factor for perinatal complications a logistic regression analysis should be performed. Please provide it.

Response: As clarified in the response to the previous comment, tuberculosis exposure was defined based on the performance of one of the diagnostic tests of tuberculosis that was performed if relevant clinical manifestations and relevant epidemiologic factors and/or positive Tuberculin Mantoux test were positive.  The following paragraph has been added to the statistical analysis paragraph of the Materials and Methods section: "A binary logistic regression model was used to study the association between tuberculosis exposure and perinatal outcomes. The Generalized Estimation Equation binary logistic model allows adjustments for siblings in the cohort and the similarities between them".

(Materials and Methods section, lines 147-150 in the revised manuscript)

  1. Please provide a necessary number of study group individuals on the basis of power analysis.

Response: We thank the reviewer for the comment.

The following paragraph has been added to the statistical analysis paragraph of the Materials and Methods section: "According to previous studies, the incidence of cesarean delivery and very low birth weight in the general population is ~ 21.1% and 0.6% of all deliveries, respectively (25,26). Assuming the rate in the unexposed group is equivalent to that of the entire population and given α=0.05, for odds ratio of 2.513 and 7.540 the calculated power is 71% and 52% for cesarean delivery and very low birth weight, respectively". 

Reference:

  1. Sandall J, Tribe RM, Avery L, et al. Short-term and long-term effects of caesarean section on the health of women and children. Lancet. 2018;392(10155):1349-1357.
  2. Murphy SL, Mathews TJ, Martin JA, Minkovitz CS, Strobino DM. Annual Summary of Vital Statistics: 2013-2014. Pediatrics. 2017;139(6):e20163239.

(Materials and Methods section, lines 154-158 in the revised manuscript)

  1. The authors found a relation between positive Mantoux test and placental abruption as well as LBW. What is the hypothetic etiology of these relations? Please provide them in the discussion part of the study.

Response: We thank the reviewer for the comment. We have consulted with the infectious disease department of our institution regarding the diagnostic tools of tuberculosis of our cohort and opened several files for verification. Accordingly, we have revised the definition of women with tuberculosis. Any patient with relevant clinical manifestations (cough >2 to 3 weeks' duration, lymphadenopathy, fevers, night sweats, weight loss) and relevant epidemiologic factors (history of prior tuberculosis infection or disease, known or possible tuberculosis exposure, and/or past or present residence in or travel to an area where tuberculosis is endemic), and/or positive Tuberculin Mantoux test, was administered one of the diagnostic tests of tuberculosis. The possible diagnostic tests, isolating mycobacterium tuberculosis from bodily secretions (culture of sputum, bronchoalveolar lavage, or pleural fluid), or tissue (pleural biopsy or lung biopsy) and sputum acid-fast bacilli smear and nucleic acid amplification testing.

The following paragraph has been added to the Materials and Methods section: "The primary exposure was maternal tuberculosis diagnosed before or during the current pregnancy. Any patient with relevant clinical manifestations (cough >2 to 3 weeks' duration, lymphadenopathy, fevers, night sweats, weight loss) and relevant epidemiologic factors (history of prior tuberculosis infection or disease, known or possible tuberculosis exposure, and/or past or present residence in or travel to an area where tuberculosis is endemic), and/or positive Tuberculin Mantoux test, was administered one of the diagnostic tests of tuberculosis. The possible diagnostic tests were isolating mycobacterium tuberculosis from bodily secretions (culture of sputum, bronchoalveolar lavage, or pleural fluid), or tissue (pleural biopsy or lung biopsy) and sputum acid-fast bacilli smear and nucleic acid amplification testing were also considered diagnostic for tuberculosis (22,23)".

References:

  1. Lewinsohn DM, Leonard MK, Lobue PA et al. Official American thoracic society/infectious diseases society of America/centers for disease control and prevention clinical practice guidelines: diagnosis of Tuberculosis in adults and children. Clin Infect Dis. 2017;15;64(2):111-115.
  2. Pai M, Nicol MP, Boehme CC. Tuberculosis diagnostics: state of the art and future directions. Microbiol Spectr. 2016;4(5).

(Materials and methods section, lines 90-99 in the revised manuscript)

  1. Line 206 – “Despite previous studies showing strong association between genital tract tuberculosis and infertility (7), our study did not demonstrate such an association…” – The data on genital tract tuberculosis occurrence in the study group is missing – please provide specific data.

Response: No cases of genital tuberculosis were seen in our population. The above sentence has been added to the Results section.

(Results section, line 165 in the revised manuscript)

  1. The English language and space inserting in the text need throughout editing.

Response: The manuscript has been edited by a native English speaker and text insertions have been corrected.

We are very grateful for the provided feedback and feel that these revisions have considerably improved the quality of this paper. It is our sincere hope that those revisions will be met with approval of the Editorial Board.

Respectfully,

Shanny Sade MD, Eyal Sheiner MD PhD, Tamar Wainstock PhD, Narkis Hermon MD, Shimrit Yaniv Salem MD, Tamar Kosef MD, Talya Lanxner Battat MD, Sharon Oron MD, Gali Pariente MD

Round 2

Reviewer 2 Report

No other comments

This manuscript is a resubmission of an earlier submission. The following is a list of the peer review reports and author responses from that submission.

Round 1

Reviewer 1 Report

The authors present data from a large retrospective cohort concerning evaluation of perinatal outcome of women with tuberculosis and long-term infectious morbidity of the newborn. 

In total n=46 deliveries in women with tuberculosis were identified during the period 1991-1994 in a single-center database. 

Maternal tuberculosis was found to be associated with placental abruption, a higher chance of ceasarean delivery and low birth weight. In a long-term perspective newborn of women with tuberculosis are not at an increased risk for infectious morbidity.

The manuscript fullfills all scientific standards and adds additional information on a relevant topic of neonatal morbiditiy, especially in low-income countries. Especially the long follow-up (over 20 years) must be highlighted. 

On the other hand, as already mentioned by the authors, the main limitation is the retrospective design. Moreover the low number of cases with n=46 limits the statistical interpretation concerning outcome and outcome-associated factors. I would recommend connecting data in sense of a multi-center database analysis to improve statistical interpretation. 

There are some minor revisions needed in regard to English language text editing (e.g. through a professional editor). 

Author Response

Careful consideration was given to the thoughtful comments of the reviewers. Each point raised received a response by revising our manuscript. We hereby state that all authors read and approved the revised version of the paper. Following are our responses to the comments made by the reviewer:

Referee 1:

The main limitation is the retrospective design. Moreover the low number of cases with n=46 limits the statistical interpretation concerning outcome and outcome-associated factors. I would recommend connecting data in sense of a multi-center database analysis to improve statistical interpretation.

Response: According to the comment of the reviewer, the following paragraph has been added to the limitation paragraph of the Discussion: "Another limitation of the study is related to the low number of mothers with tuberculosis, which may limit the statistical interpretation concerning outcome and outcome- associated factors. Extending the timeline of the study to capture more patients or conducting the same study in other region with higher incidence might have strengthened the results of this study, as the significant finding affected only a low number of pregnancies. Further studies using a multi-center database analysis should be done in order to further investigate perinatal outcome of women with known tuberculosis". (Discussion section, lines 244-250 in the revised manuscript).

There are some minor revisions needed in regard to English language text editing (e.g. through a professional editor).

Response: English language editing has been done by a native English speaker.

Reviewer 2 Report

Many thanks for the invitation to review the manuscript entitled “Perinatal outcome and long-term infectious morbidity of offspring born to women with known tuberculosis”. I enjoyed reading this manuscript and appreciate the authors’ use of an extremely large perinatal and pediatric database to achieve their objectives. My specific comments to the authors are below:

I fear that the number of patients with TB (46) is too low to draw useful inferences regarding both perinatal complications and also infectious related morbidity, which for many of these have an extremely low baseline incidence.

The study should have either been extended in its timeline to capture more patients or been conducted in other regions/hospitals with higher incidence. The author’s themselves state that worldwide, 216,000 women have TB during pregnancy, therefore only a tiny fraction of them are included here.

Regarding the significant findings of increased abruption (6.5%) and very low birthweight (4.3%) in mothers with TB, clinically-speaking, these only affected 3 and 2 pregnancies respectively. Although the statistics support there being a difference, rationally this may still be due to chance alone. Controlling for maternal and gestational age is certainly a useful thing to do however it is asking too much of the small TB positive data set.

Author Response

Careful consideration was given to the thoughtful comments of the reviewers. Each point raised received a response by revising our manuscript. We hereby state that all authors read and approved the revised version of the paper. Following are our responses to the comments made by the reviewer:

Referee 2:

I fear that the number of patients with TB (46) is too low to draw useful inferences regarding both perinatal complications and also infectious related morbidity, which for many of these have an extremely low baseline incidence.

The study should have either been extended in its timeline to capture more patients or been conducted in other regions/hospitals with higher incidence. The author’s themselves state that worldwide, 216,000 women have TB during pregnancy, therefore only a tiny fraction of them are included here.

Response: According to the comment of the reviewer, the following paragraph has been added to the limitation paragraph of the Discussion: "Another limitation of the study is related to the low number of mothers with tuberculosis, which may limit the statistical interpretation concerning outcome and outcome- associated factors. Extending the timeline of the study to capture more patients or conducting the same study in other region with higher incidence might have strengthened the results of this study, as the significant finding affected only a low number of pregnancies. Further studies using a multi-center database analysis should be done in order to further investigate perinatal outcome of women with known tuberculosis". (Discussion section, lines 244-250 in the revised manuscript).

Regarding the significant findings of increased abruption (6.5%) and very low birthweight (4.3%) in mothers with TB, clinically-speaking, these only affected 3 and 2 pregnancies respectively. Although the statistics support there being a difference, rationally this may still be due to chance alone. Controlling for maternal and gestational age is certainly a useful thing to do however it is asking too much of the small TB positive data set

Response: According to the comment of the reviewer, the following paragraph has been added to the limitation paragraph of the Discussion: "The significant findings of increased abruption and very low birthweight in mothers with tuberculosis only affected 3 and 2 pregnancies respectively. Nevertheless, it is noteworthy that the results remained significant even after controlling for maternal age and gestational age. Although the statistics support there being a difference, rationally this may still be due to chance alone".

(Discussion section, lines 250-254 in the revised manuscript).

Reviewer 3 Report

This study describes the pregnancy complications of maternal tuberculosis and offspring follow-up in a large single center cohort in Israel. Earlier known complications were confirmed in a group of 46 women and no long term infectious morbidity for the offspring up to 18 years was found. The first finding is not new, but long term follow-up is always difficult to do and a relevant topic. The authors themselves however also raise the limitations of this study.

  1. Quality of the follow–up is always a problem, were all 46 children retrieved for follow-up?
  2. The study period in years in which the 243,682 deliveries took place was from 1991-2014, So from around 10,595 deliveries a year, 46 proven cases in 23 years, that is 2 cases a year. How were they diagnosed? How many missed? Table 1: 2.2 % IVF with known TBC?
  3. What is the referral policy from other hospitals to this third-level institute? Is there a selection bias?
  4. Were missing diagnosis of TBC in the control group considered, or is there a general screening policy?
  5. table 2: perinatal mortality 2.2 % of 46 is 1 case (of which of course, no follow up is reported) with these small numbers giving percentages is not useful and confidence intervals are large, just show the numbers. Same for placental abruption, 3 cases from the 46, one death, two survived? At what gestational ages? Only cases with active TBC? With 46 cases it should be not much work to fully analyse and report on these cases from the retrieved clinical notes.

In summary, this paper contains some interesting data but there are too many questions for publishing it in this format.

Author Response

Careful consideration was given to the thoughtful comments of the reviewers. Each point raised received a response by revising our manuscript. We hereby state that all authors read and approved the revised version of the paper. Following are our responses to the comments made by the reviewer:

Referee 3:

  1. Quality of the follow–up is always a problem, were all 46 children retrieved for follow-up?

Response: According to the comment of the reviewer, the following sentence has been added to the Results section: "All children were retrieved for follow-up except from one case of ante-partum fetal death".

(Results section, lines 159-160 in the revised manuscript).

  1. The study period in years in which the 243,682 deliveries took place was from 1991-2014, so from around 10,595 deliveries a year, 46 proven cases in 23 years, that is 2 cases a year. How were they diagnosed?

Response: According to the comment of the reviewer, the following sentence has been added to the study design paragraph of the Materials and methods section: "The primary exposure was maternal tuberculosis diagnosed before or during the current pregnancy by a Tuberculin Mantoux test". (Materials and methods section, lines 76-77 in the revised manuscript).

How many missed?

Response: According to the comment of the reviewer, the following paragraph has been added to the limitation paragraph of the Discussion: "Lastly, since there is no general screening in Israel for tuberculosis, differential misclassification of the study exposure (cases with tuberculosis) may have appeared in the comparison group (unexposed group). Meaning among the unexposed group there were actually undiagnosed cases of tuberculosis. This, however, may have led to an underestimation of the true association between tuberculosis with adverse perinatal outcome, found in our study".

(Discussion section, lines 254-258 in the revised manuscript).

Table 1: 2.2 % IVF with known TBC?

Response: According to the comment of the reviewer, the following sentence has been added to the Results section: "No differences were found in the in- vitro fertilization rates between the two groups". (Results section, line 134 in the revised manuscript).

According to the comment of the reviewer, the following paragraph has been added to the Discussion section:

"Despite previous studies showing strong association between genital tract tuberculosis and infertility (7), our study did not demonstrate such an association, although in-vitro fertilization procedures in Israel are not performed in women with active tuberculosis". 

(Discussion section, lines 202-204 in the revised manuscript)

  1. What is the referral policy from other hospitals to this third-level institute? Is there a selection bias?

Response: According to the comment of the reviewer, the following sentence has been added to the strengths paragraph of the Discussion: "Our hospital is the only hospital serving the entire population of the region. The hospital provides both maternal and pediatric services, hence enables to combine databases from obstetrical and pediatric departments, from pregnancy to adulthood. These facts allow low probability of selection bias".

(Discussion section, lines 233-236 in the revised manuscript)

  1. Were missing diagnosis of TBC in the control group considered, or is there a general screening policy?

Response: According to the comment of the reviewer, the following sentence has been added to the limitations paragraph of the Discussion: ""Lastly, since there is no general screening in Israel for tuberculosis, differential misclassification of the study exposure (cases with tuberculosis) may have appeared in the comparison group (unexposed group). Meaning among the unexposed group there were actually undiagnosed cases of tuberculosis. This, however, may have led to an underestimation of the true association between tuberculosis with adverse perinatal outcome, found in our study".  

(Discussion section, lines 254-258 in the revised manuscript).

  1. Perinatal mortality 2.2 % of 46 is 1 case (of which of course, no follow up is reported) with these small numbers giving percentages is not useful and confidence intervals are large, just show the numbers. Same for placental abruption, 3 cases from the 46, one death, two survived? At what gestational ages? Only cases with active TBC? With 46 cases it should be not much work to fully analyse and report on these cases from the retrieved clinical notes.

Response: According to the comment of the reviewer, numbers were added to table 2.

 The following sentence has been added to the Results section: "There were 3 cases of placental abruption of which one case of perinatal mortality. The only perinatal mortality was an ante- partum death case at term in a woman with latent tuberculosis ".

(Results section, lines 143-145 in the revised manuscript)

Round 2

Reviewer 3 Report

Having read the responses of the authors to the concerns of the reviewers, I do not see a substantial improvement of this paper enough to warrant publication.

The addition of the tuberculin Mantoux test is puzzling, there seems to be no general screening policy, so on what indication has this test been perfomed in this cohort, and a Mantoux is also positive afte BCG vaccination.